

# Identification of diagnostic genes and vital microRNAs involved in rheumatoid arthritis: based on data mining and experimental verification

Conglin Ren[1,*], Mingshuang Li[2,*], Yang Zheng[1], Fengqing Wu[1], Weibin Du[3] and Renfu Quan[1,3]

[1] The Third Clinical Medical College of Zhejiang Chinese Medical University, Hangzhou, Zhejiang, China
[2] The First Affiliated Hospital of Zhejiang Chinese Medical University, Hangzhou, Zhejiang, China
[3] Department of Orthopedics, Xiaoshan Traditional Chinese Medicine Hospital, Hangzhou, Zhejiang, China
[*] These authors contributed equally to this work.

## ABSTRACT

**Background.** The pathogenesis of rheumatoid arthritis (RA) is complex. This study aimed to identify diagnostic biomarkers and transcriptional regulators that underlie RA based on bioinformatics analysis and experimental verification.

**Material and Methods.** We applied weighted gene co-expression network analysis (WGCNA) to analyze dataset GSE55457 and obtained the key module most relevant to the RA phenotype. We then conducted gene function annotation, gene set enrichment analysis (GSEA) and immunocytes quantitative analysis (CIBERSORT). Moreover, the intersection of differentially expressed genes (DEGs) and genes within the key module were entered into the STRING database to construct an interaction network and to mine hub genes. We predicted microRNA (miRNA) using a web-based tool (miRDB). Finally, hub genes and vital miRNAs were validated with independent GEO datasets, RT-qPCR and Western blot.

**Results.** A total of 367 DEGs were characterized by differential expression analysis. The WGCNA method divided genes into 14 modules, and we focused on the turquoise module containing 845 genes. Gene function annotation and GSEA suggested that immune response and inflammatory signaling pathways are the molecular mechanisms behind RA. Nine hub genes were screened from the network and seven vital regulators were obtained using miRNA prediction. CIBERSORT analysis identified five cell types enriched in RA samples, which were closely related to the expression of hub genes. Through ROC curve and RT-qPCR validation, we confirmed five genes that were specific for RA, including CCL25, CXCL9, CXCL10, CXCL11, and CXCL13. Moreover, we selected a representative gene (CXCL10) for Western blot validation. Vital miRNAs verification showed that only the differences in has-miR-573 and has-miR-34a were statistically significant.

**Conclusion.** Our study reveals diagnostic genes and vital microRNAs highly related to RA, which could help improve our understanding of the molecular mechanisms underlying the disorder and provide theoretical support for the future exploration of innovative therapeutic approaches.

Corresponding author
Renfu Quan, quanrf@yeah.net

## INTRODUCTION

Rheumatoid arthritis (RA) is a disorder characterized by symmetrical, aggressive joint inflammation (*Smolen, Aletaha & McInnes, 2016*). The prevalence of RA varies somewhat by ethnicity and geographic region. According to the literature, African Americans have a higher prevalence of the disease by approximately 1.02%, compared to Hispanic Americans, who have a much lower risk of this disease at only 0.45% (*Kawatkar et al., 2012*). The etiology of RA is unclear, and current research suggests that it may result from a combination of factors, including genetics, infection, and sex hormones. A study on twins showed that the genetic probability of RA is approximately 60% and is not influenced by sex, age of onset, or disease severity (*MacGregor et al., 2000*). The changes in the composition and function of gut microbes are closely related to infection by autoimmune diseases. The microbial diversity of the gut is significantly reduced in RA patients compared to healthy individuals, and the proportion of certain rare bacteria like Actinobacteria is elevated (*Chen et al., 2016*). More interestingly, oral infections whose clinical manifestations seem unrelated to RA have also been identified as risk factors in recent studies (*Hajishengallis, 2015*; *Kharlamova et al., 2016*). The pathogenesis of RA is complex and clinical symptoms vary considerably between individuals. There have been no reliable biomarkers identified to date, despite the discovery of multiple susceptibility genes associated with RA (e.g., IL2RA and MMP9) and new advances in the development of therapeutic drugs.

microRNA (miRNA) has become a key node in biomedical research as an important factor regulating gene transcription and post-transcriptional regulation. Numerous findings support the idea that miRNAs are essential for maintaining immune system homeostasis, including the regulation of T-cell activation, immune tolerance and the inflammatory response, and the aberrant expression of miRNAs may lead to increased susceptibility to autoimmune diseases (*O'connell et al., 2010*; *Simpson & Ansel, 2015*). Recently, the deregulation of endogenous miR-155 has been shown to be involved in malignant tumorigenesis and in abnormal autoimmune conditions such as RA and systemic sclerosis (*Leng et al., 2011*). Using miRNA profiling, *Pan et al. (2010)* found that miR-21 and miR-148a were highly expressed in T cells from lupus patients and animal models, which inhibited DNMT1 gene expression to induce cellular hypomethylation (*Pan et al., 2010*). Therefore, an in-depth study of the functional links between miRNAs and mRNAs may reveal the pathogenesis of diseases and develop new therapeutic approaches.

With the advent of microarray and high-throughput sequencing technologies, bioinformatics is becoming a popular discipline in biomedical research, and analysis tools are constantly being developed and upgraded. Weighted gene co-expression network analysis (WGCNA) is an algorithm for detecting correlation between genes (*Langfelder & Horvath, 2008*; *Langfelder & Horvath, 2012*), which facilitates network-based genetic screening methods and is helpful to uncover potential biomarkers and therapeutic
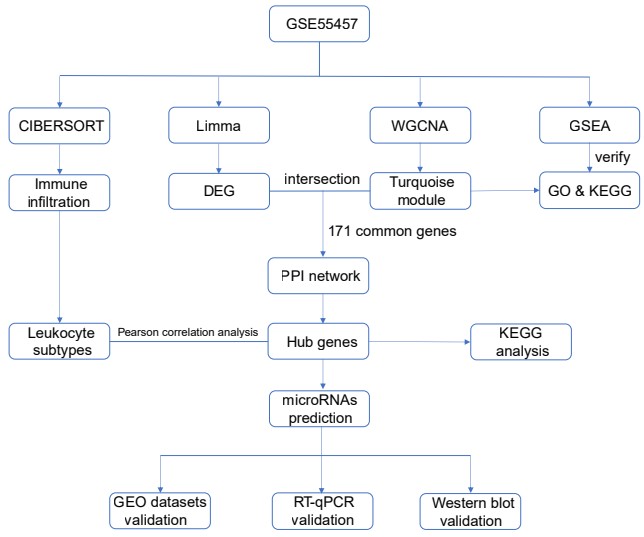

**Figure 1    Flow chart of the study design.**

targets for diseases. WGCNA has been successfully practiced in studies of autoimmune diseases, such as juvenile dermatomyositis (*Zhong et al., 2020*), lupus nephritis (*Yang & Li, 2019*), ulcerative colitis (*Zhang et al., 2020*) and Sjögren's syndrome (*Yao et al., 2019*). We integrated multiple bioinformatics methods to analyze microarray datasets derived from the Gene Expression Omnibus (GEO) repository and validated the data mining results in three ways in order to identify diagnostic genes and transcriptional regulators of RA. Figure 1 summarized the entire process of our research.

## MATERIAL AND METHODS

### Data download and processing

Gene expression matrices and the clinical information of GSE55457 based on platform GPL96 [HG-U133A], were downloaded from the GEO database. Twenty-three samples were selected for our study. We performed normalization and log2 transformation on original expression matrices and used a platform annotation file to convert probes into gene symbols. If a gene was detected by more than one probe, the average expression value was used for subsequent analysis.

Differentially expressed genes (DEGs) between RA patients and healthy controls were analyzed by limma, which is a tailored R package for expression profiling studies (*Ritchie et al., 2015*). The DEGs filtering thresholds were set to |logFC|>1 and the adjusted *p*-value was < 0.05.

### Weighted correlation network analysis

Gene co-expression analysis of dataset GSE55457 was performed in R software (version: 3.6.3) based on the WGCNA R package reference manual. We calculated the median absolute deviation (MAD) for each gene, which reflected the degree of variation in gene expression across samples, and selected the top 3,000 highly variable genes for network

construction and modules detection. First, we clustered samples to detect the presence of outliers. Next, optimal soft-thresholding power ($\beta$) was selected, which was a pivotal step in the entire analysis process. In order to reduce interference from spurious correlations, the adjacency matrix was replaced by a topological overlap matrix (TOM), and then the hclust function was used to generate a hierarchical clustering tree (dendrogram) of genes. The detailed parameters used for modules detection were as follows: minModuleSize was set to 30 to obtain large modules; modules with a correlation of 0.75 were merged using mergeCloseModules function. We also quantified module–trait associations and their $p$-values, which allowed us to pick out modules that were closely related to the trait of interest for next step analysis. The absolute value of correlation coefficient $> 0.6$ and $p$-value $< 0.05$ were cutoffs for module screening. Finally, we measured the gene significance (GS) and module membership (MM) of individual genes and scatterplots were drawn.

## Functional annotation and GSEA

R package clusterProfiler (*Yu et al., 2012*) and its dependency annotation package org.Hs.eg.db, sourced from Bioconductor, were used to analyze and visualize the functional profiles (GO and KEGG) of key module genes. To obtain highly credible analysis results, only enrichment items satisfying both $p$-value $< 0.05$ and $q$-value $< 0.05$ were outputted. Moreover, gene set enrichment analysis (GSEA) (*Mootha et al., 2003*; *Subramanian et al., 2005*), which can be used to determine whether an already-defined gene set exhibits a statistically significant difference in two different traits (e.g., RA patients and healthy individuals), was conducted to explore biological processes and pathways that may be involved in the pathogenesis of RA. The complete gene expression matrix and phenotype information of dataset GSE55457 were uploaded to GSEA and analyzed with hallmark gene sets that are stored in molecular signatures database (MSigDB) (*Liberzon et al., 2015*; *Liberzon et al., 2011*). Gene sets enriched at FDR $< 0.25$ and $p$-value $< 0.01$ were retained.

## Building PPI network and mining hub genes

The intersection of DEGs and key module genes was entered into the STRING database to find interrelationships (*Szklarczyk et al., 2015*; *Szklarczyk et al., 2019*; *Szklarczyk et al., 2017*). Due to the large size of the initial protein-protein interaction (PPI) network, we hid isolated nodes and limited the minimum interaction score to 0.7. Then, the network was exported and further optimized in Cytoscape software (*Cline et al., 2007*; *Otasek et al., 2019*; *Shannon et al., 2003*). The MCODE plugin was installed in Cytoscape to find clusters made up of highly related genes in the network (*Bader & Hogue, 2003*). The parameters used in our analysis were default values. Genes in the highest scoring cluster were considered to be hub genes of RA and were re-performed for KEGG analysis to explore vital biological pathways.

## Estimating proportion of immune cells in synovial tissue

Understanding the immune microenvironment in RA may provide new insights into the immunotherapy of the disease. The CIBERSORT tool used for estimating the immune cell contents of different samples in a gene expression admixture has a wide range of applicability and high sensitivity, which is important for disease diagnosis and therapeutic

target predictions (*Newman et al., 2015*). A mixture file was generated according to the input format requirements, and we selected default LM22 as signature gene file to distinguish the hematopoietic populations and activation states. The analysis results were filtered based on a threshold of *p*-value < 0.05 to remove samples with a low confidence level. We conducted correlation analysis between the hub genes and immune cell subtypes significantly enriched in RA samples according to Pearson's algorithm.

## Prediction of gene-miRNA relationships

MiRNAs are a class of non-coding RNAs that regulate gene expression by making mRNA unstable and inhibiting translation, a process by which mRNA is converted into protein. Here, MCODE-identified hub genes were uploaded to the online analysis tool miRDB; we searched for miRNAs that have regulatory roles in the development of RA (*Chen & Wang, 2020*; *Liu & Wang, 2019*). The parameters were set as follows: only functional miRNAs were considered; gene targets with a score below 60 and miRNAs with more than 2,000 targets were excluded. Cytoscape was then used to construct a gene-miRNA network in which miRNAs targeting two or more genes were considered to be vital regulators involved in RA pathogenesis.

## Validating hub genes and miRNAs with independent microarray datasets

The expression matrix GSE12021 was selected as the validation dataset for hub genes. We plotted ROC curves in R software based on the pROC package to test whether screened hub genes have good diagnostic value for RA (*Robin et al., 2011*). The area under the curve (AUC) was computed according to the trapezoidal rule and genes with AUC > 85% were further validated by RT-qPCR. To understand the differences in transcriptional regulators between RA patients and healthy individuals, we validated vital miRNAs with GEO dataset GSE37425, which focused on the expression of miRNAs in synovial tissue using the TaqMan probe approach. The entire gene expression matrix was normalized and then transformed with log2.

## Validating hub genes and miRNAs with RT-qPCR

Five RA patients and three healthy individuals who attended the health examination center of Xiaoshan Traditional Chinese Medicine Hospital were included and all subjects signed consent forms. This study was approved by Ethics Committees of Xiaoshan Traditional Chinese Medicine Hospital (NO. 2020012). We collected blood samples from each test subject and isolated the peripheral blood mononuclear cells (PBMCs) using human lymphocyte separation medium (Solarbio, China). Next, the total RNA of PBMCs was extracted using the TRIzol reagent (Invitrogen, USA) and its purity was assessed by NanoDrop (Thermo Scientific, USA). We used a cDNA synthesis kit (Thermo Scientific, USA) to generate the first strand cDNA and cDNA amplification was performed on a 7500 real-time PCR instrument (Applied Biosystems, USA). For the quantitative analysis of miRNA expression, Hairpin-it miRNAs qPCR quantitation kit (Genepharma, China) was applied. We chose GAPDH and U6 as internal reference genes and compared relative

**Table 1  RT-qPCR primers used in this study.**

| Gene symbol | Sequence (5′ to 3′) |
| --- | --- |
| CCL25 | forward: TATTCTACCTCCCCAAGAGACA<br>reverse: GATGGGATTGCTAAACTTGGAC |
| CXCL10 | forward: CTCTCTCTAGAACTGTACGCTG<br>reverse: ATTCAGACATCTCTTCTCACCC |
| CCL5 | forward: CAGCAGTCGTCCACAGGTCAAG<br>reverse: TTTCTTCTCTGGGTTGGCACACAC |
| CXCL9 | forward: AAGACCTTAAACAATTTGCCCC<br>reverse: TGCTGAATCTGGGTTTAGACAT |
| CXCL13 | forward: CAAGGTGTTCTGGAGGTCTATT<br>reverse: TGAATTCGATCAATGAAGCGTC |
| CXCL11 | forward: GCTGTGATATTGTGTGCTACAG<br>reverse: TTGGGTACATTATGGAGGCTTT |
| GAPDH | forward: CGGACCAATACGACCAAATCCG<br>reverse: AGCCACATCGCTCAGACACC |
| miR-34a-5p | forward: TCTGTCTCTCTTGGCAGTGTCTTA<br>reverse: AATGGTTGTTCTCCACTCTCTCTC |
| U6 | forward: CAGCACATATACTAAAATTGGAACG<br>reverse: ACGAATTTGCGTGTCATCC |

mRNA or miRNA expression based on the $2^{-\Delta\Delta CT}$ method. The primer sequences are available in Table 1.

## Western blot

Total proteins were extracted from PBMCs using RIPA lysis buffer (Beyotime Biotechnology, China), followed by concentration detection with the BCA protein assay kit (Sangon Biotech, China). Equal amounts of protein were separated by 8–20% SDS-PAGE and transferred onto polyvinylidene difluoride membranes. Afterwards, membranes were blocked with 5% skim milk for 1 h and then incubated with primary antibodies including $\beta$-actin (1:1000, Cell Signaling Technology (CST), #3700), and CXCL10 (1:1000, CST, #14969) at 4 °C overnight. Anti-mouse IgG (1:2000, CST, #7076) and anti-rabbit IgG (1:2000, CST, #7074) were used to bind the primary antibodies. Finally, membranes were scanned using FluorChem Q (Proteinsimple, USA). Image J software was used for the quantitative analysis of protein bands.

## Statistical analysis

Statistical analysis was performed in GraphPad Prism 7.0. Data satisfying normal distribution were analyzed by unpaired $t$-test; the Mann–Whitney test was used for data that were not normalized. $P$-value < 0.05 was considered significant.

## RESULTS

### Screening for DEGs

Our differential expression assessment of dataset GSE55457 identified 367 DEGs, of which 186 genes were highly expressed and 181 genes had low expression levels in RA patients. We created a volcano map and heat map for DEGs as shown in Fig. 2.

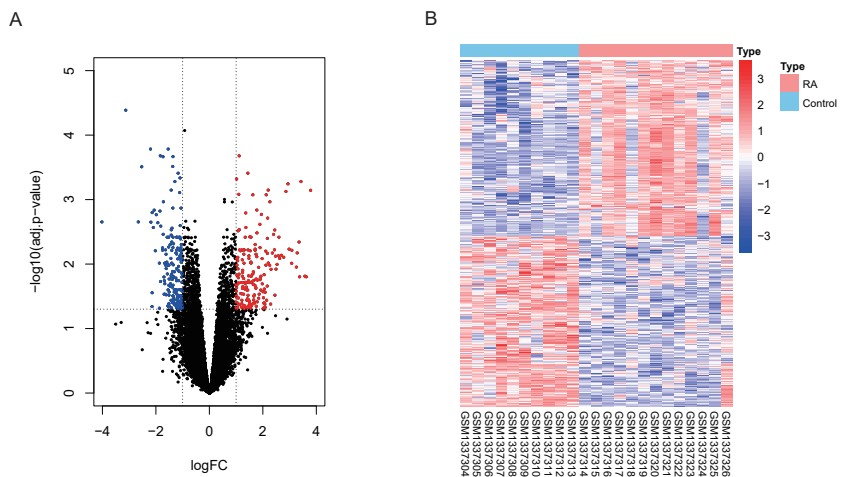

**Figure 2  Visualization of differentially expressed genes (DEGs).** (A) DEGs filtered by thresholds were presented in volcano map. Red dots represent up-regulated genes and blue dots represent down-regulated genes. (B) Heatmap showed the expression of DEGs in each sample.

## Identification of key module closely related to RA

Two samples (GSM1337307 and GSM1337308) were identified as outliers and were excluded by the cutreeStatic function with parameter cutHeight = 90 (Fig. 3A). The soft-thresholding power ($\beta$) was set to 5, for which the scale-free topology fit index ($R^2$) reached 0.90 (Fig. 3B). Co-expression analysis resulted in 14 modules, and the size of each module is shown in Figs. 3C and 3D. Figure 3E showed the relationships between modules and traits, with four modules (black, turquoise, blue, and purple) having strong positive or negative correlations with disease status (presence or absence of RA). Taking into account gene numbers, correlation coefficients, and $p$-values of the above four modules, we focused on the turquoise module containing 845 genes and performed an in-depth analysis. The GS and MM plot of the turquoise module (correlation coefficient = 0.79, $p$-value = 2.7e−181) demonstrated that the closer the gene is to the trait, the more important it is in the module (Fig. 3F).

## Functional annotation of turquoise module genes and GSEA

The top 10 results for GO enrichment are shown in Fig. 4A and mainly involved leukocyte activation and cell adhesion. To associate a specific gene with its assigned GO terms, we adopted the GOChord function to generate a circle diagram (Fig. 4B). Figure 4C pointed out that inflammation-related signaling pathways, such as chemokine signaling pathway, NF-kappa B signaling pathway, Th17 cell differentiation and T cell receptor signaling pathway, were significantly linked to the pathogenesis of RA. Similarly, a circle plot was displayed to visualize the correspondence between genes and pathways (Fig. 4D). We identified five gene sets based on a reference database derived from MSigDB using the GSEA method. These were associated with the immune response or inflammatory pathways that were highly enriched in RA, which verified the GO and KEGG results (Figs. 5A–5E).

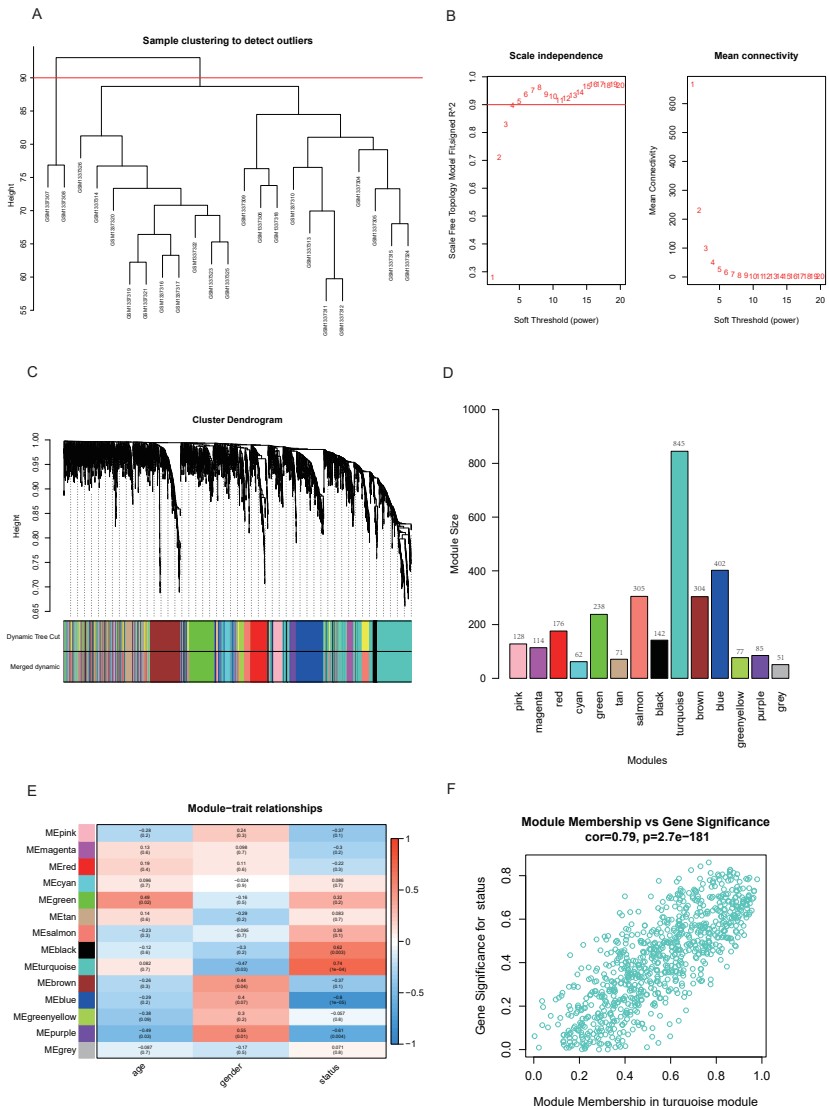

**Figure 3 Identification of key module closely related to RA.** (A) Clustering of samples and removal of outliers. (B) $\beta = 5$ was chosen as appropriate soft-thresholding power, for which scale-free topology fit index ($R^2$) reached 0.90. (C) Co-expression analysis resulted in 14 modules. Each branch represents one gene, and every color represents a co-expression module. (D) Number of genes in each co-expression module. (E) Heatmap showed the correlation between module eigengenes and clinical traits. (F) Gene significance (GS) and module membership (MM) of each gene in turquoise module.

## Constructing PPI network to find hub genes

A total of 171 intersection genes were entered into the STRING database to establish the network (Fig. 6A). The network was adjusted using Cytoscape software, resulting in 52 nodes and 169 relationship pairs. The MCODE plugin uncovered five clusters, with the highest scoring cluster containing nine nodes, which were potential hub genes for RA (Fig. 6B, Table 2). Furthermore, we re-performed KEGG analysis on the nine genes to investigate the signaling pathways in which they were involved. As shown in Fig. 6C, we

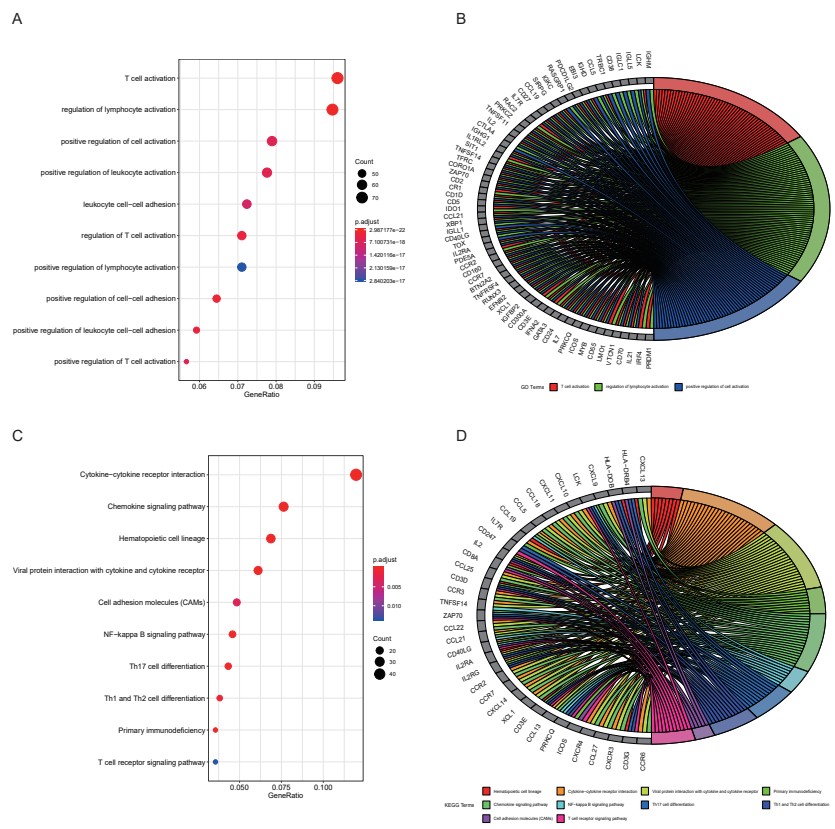

**Figure 4** **Results of functional enrichment analysis.** (A) GO analysis of turquoise module genes. (B) Circle plot was displayed to visualize correspondence between genes and top three GO terms. (C) KEGG analysis of turquoise module genes. (D) Circle plot was displayed to visualize correspondence between genes and top 10 KEGG pathways.

**Table 2** **Detailed information of each cluster.**

| Cluster | Score | Nodes | Edges | Node IDs |
|---------|-------|-------|-------|----------|
| 1 | 9.000 | 9 | 36 | CCL25, CCL5, CXCL10, CXCL11, CXCL13, CXCL9, CXCR6, GPR18, PNOC |
| 2 | 6.286 | 8 | 22 | BLNK, CD19, CD72, CD79A, CD79B, IGLL5, ITK, PLCG2 |
| 3 | 4.400 | 6 | 11 | CD8A, MAP4K1, CD247, TRAT1, CD3D, GZMA |
| 4 | 4.000 | 4 | 6 | BIRC3, PSMB10, PSMB9, TNFSF11 |
| 5 | 3.000 | 3 | 3 | CD2, CD48, CD52 |

found that nine hub genes played important roles in inflammatory or immune-related pathways, including the chemokine signaling pathway, toll-like receptor signaling pathway, and TNF signaling pathway.

## Immune infiltration and Pearson correlation analysis

One sample (GSM1337307) was excluded from analysis results because its *p*-value did not meet the threshold we set. The contents of various immune cells in each sample were

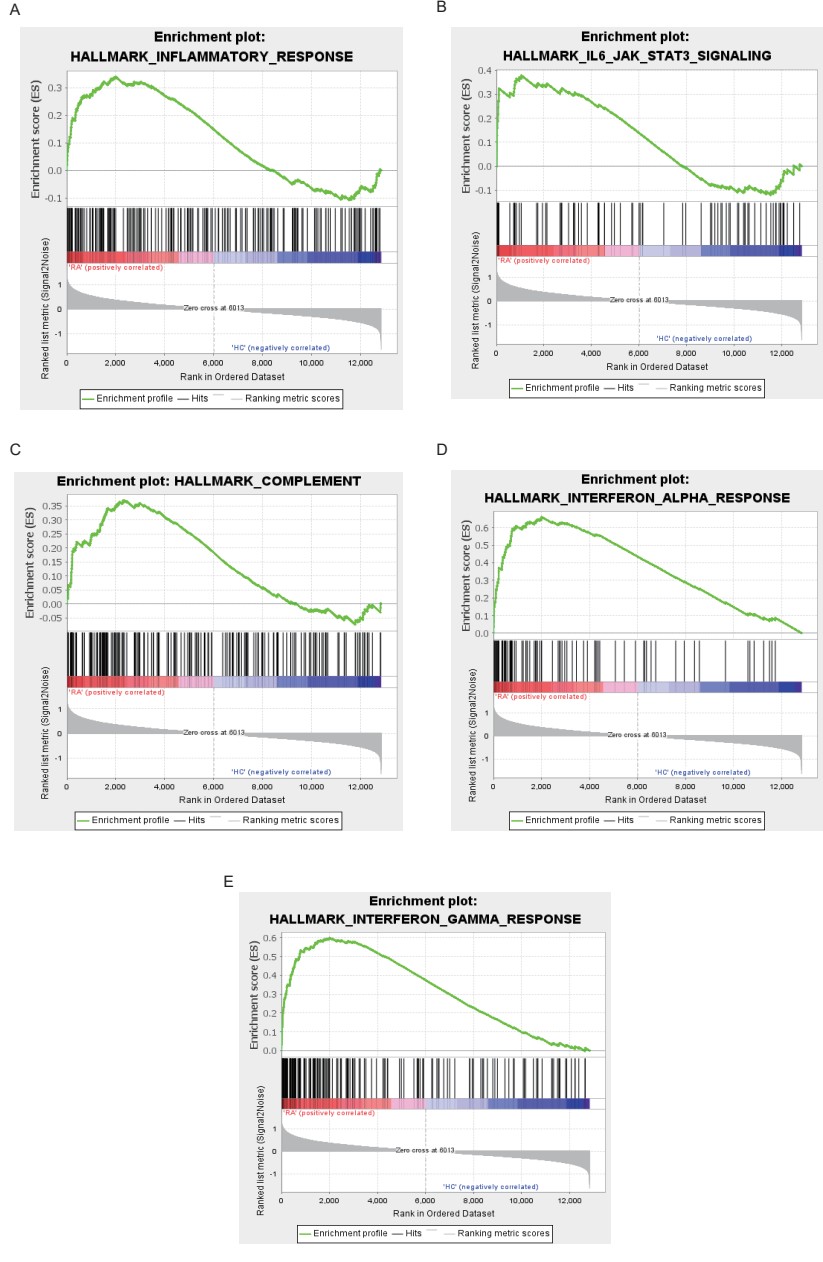

**Figure 5  Results of GSEA analysis.** Enrichment plots for inflammatory response (A), IL6-JAK-STAT3 signaling (B), complement (C), interferon alpha response (D), interferon gamma response (E).

presented by stacked bar chart (Fig. 7A). M2 macrophages accounted for the highest proportion in both RA and normal tissue. We conducted a Wilcoxon test to further clarify whether the level of each immune cell type was statistically different between two groups. From the violin plot (Fig. 7B), we inferred that five cell types, including plasma cells, CD8 T cells, follicular helper T cells, $\gamma\delta$ T cells, and M1 macrophages, were abundant in RA synovial tissue compared to the normal group. More importantly,

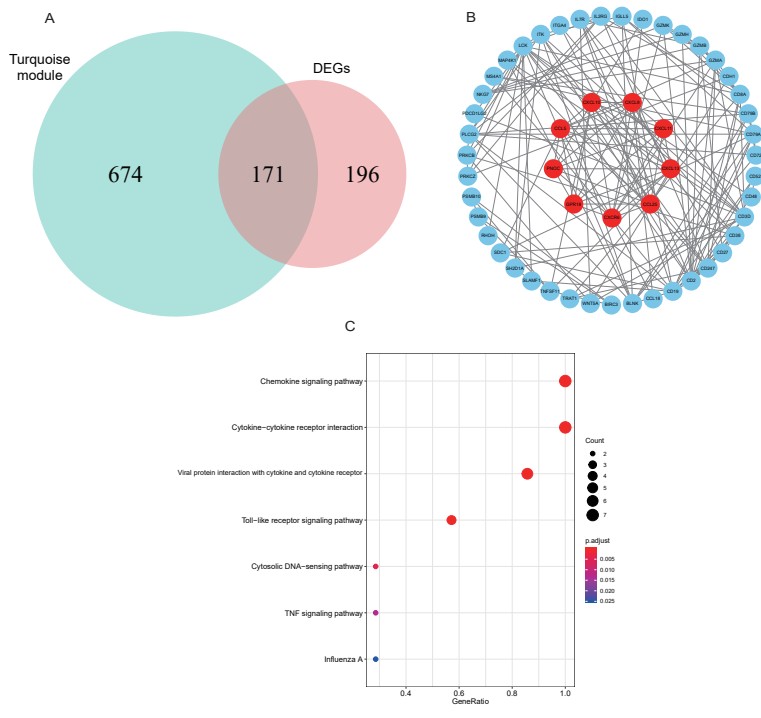

**Figure 6** **PPI network construction and mining hub genes.** (A) The intersection of DEGs and turquoise module genes was presented in Venn diagram. (B) The PPI network of intersection genes was constructed in Cytoscape. Genes in the highest scoring cluster were marked in red. (C) KEGG analysis of nine hub genes.

Pearson correlation analysis confirmed that, in addition to CCL25, eight other hub genes identified by MCODE played pro-inflammatory roles. Their overexpression may recruit large numbers of inflammatory cells to lesion sites, which is a critical pathological basis for RA (Fig. 7C).

## Predicting miRNAs that regulate nine hub genes

Normally, miRNAs can fine-tune gene expression to maintain the homeostasis of life, while abnormal miRNAs expression is thought to be associated with diseases, including benign or malignant tumors, and immune system dysfunction. Our prediction of miRNAs was made in the miRDB database and 72 miRNA-gene regulatory relationships were obtained. We then created an interactive network in Cytoscape and analyzed the number of genes targeted by each miRNA (Fig. 8). The miRNAs possessing two target genes are listed in Table 3 and were vital regulators in the pathogenesis of RA.

## Validation of hub genes by ROC curves

The expression matrix of dataset GSE12021 was chosen to test the diagnostic efficacy of hub biomarkers for RA. Probability curves (ROC curves) were plotted and the AUC reflected the degree or measure of separability. A higher AUC indicated that the gene is better at discriminating between diseased and non-diseased individuals. As shown in Fig. 9, six genes had AUC values greater than 85%, including CCL25, CXCL10, CCL5, CXCL9, CXCL13,

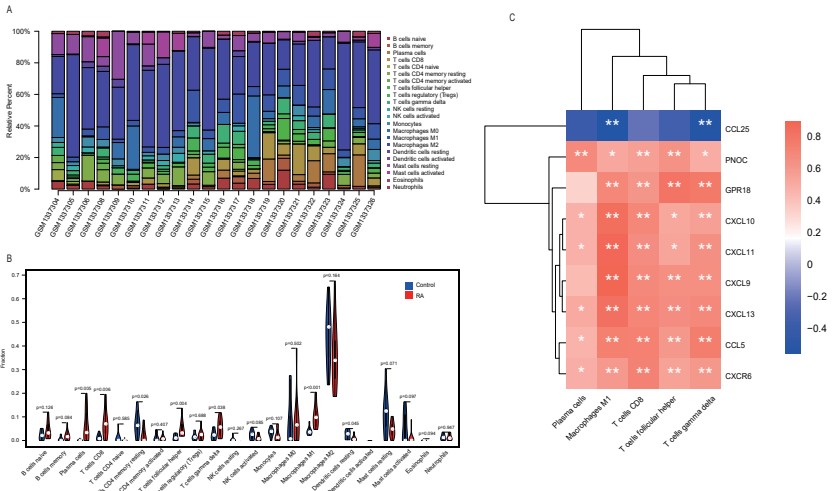

**Figure 7** **Immune infiltration analysis performed by CIBERSORT.** (A) Stacked bar chart showed contents of various immune cells in each sample. (B) The contents of 22 types of immune cells in normal (blue color) and RA (red color) groups were compared. *P*-value < 0.05 was considered statistically significant. (C) Heat map of correlation between hub genes and leukocyte subtypes. The shades of color represent the magnitude of correlation coefficient. * *p*-value < 0.05; ** *p*-value < 0.01.

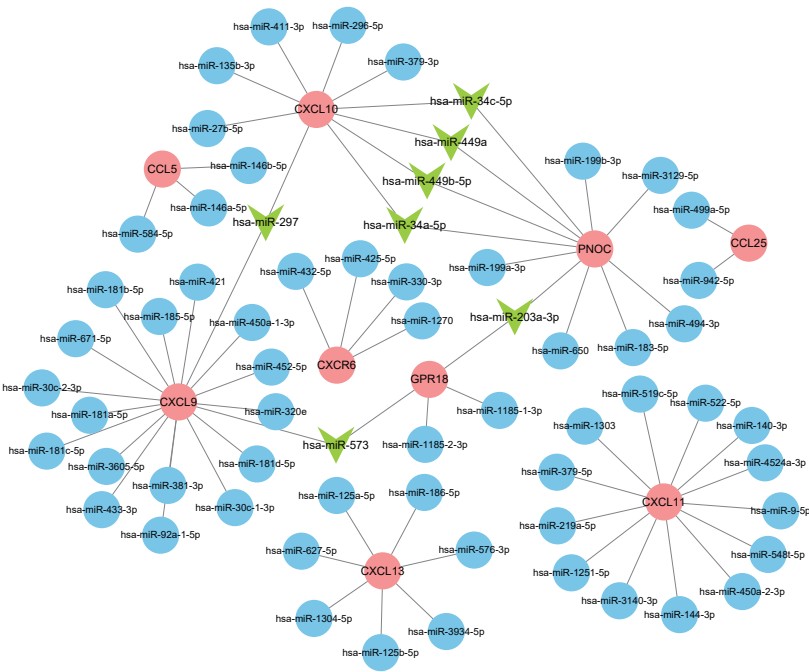

**Figure 8** **Interaction network between hub genes and miRNAs.** Hub genes were colored in red; miRNAs with one target gene were indicated by blue nodes; core miRNAs targeting two genes were indicated by green arrows.

**Table 3  Vital miRNAs and its target genes.**

| miRNA | Target genes | Count |
|---|---|---|
| hsa-miR-297 | CXCL9, CXCL10 | 2 |
| hsa-miR-573 | CXCL9, GPR18 | 2 |
| hsa-miR-203a-3p | GPR18, PNOC | 2 |
| hsa-miR-449b-5p | PNOC, CXCL10 | 2 |
| hsa-miR-34c-5p | PNOC, CXCL10 | 2 |
| hsa-miR-449a | PNOC, CXCL10 | 2 |
| hsa-miR-34a-5p | PNOC, CXCL10 | 2 |

and CXCL11, indicating that these genes were potential biomarkers for the diagnosis of RA.

### Validation of hub genes by RT-qPCR and Western blot

We collected peripheral blood from subjects for RT-qPCR validation to further evaluate the six genes with AUC > 85% identified by ROC curves. Five genes were differentially expressed between RA patients and normal subjects (Fig. 10A). Among them, CXCL9, CXCL10, CXCL11, and CXCL13 were up-regulated genes, while CCL25 was down-regulated. Moreover, to detect the expression of genes in protein level, we chose a representative gene (CXCL10) for Western blot validation. As shown in Figs. 10B and 10C, the protein and mRNA level of gene expression were consistent.

### Vital miRNAs verification

The vital miRNAs possessing two target genes were validated with GSE37425. Among the six miRNAs investigated, has-miR-449b, has-miR-573, and has-miR-203 expression was suppressed in RA patients (Figs. 11A–11F). It is worth noting that only the differences in has-miR-573 and has-miR-34a were statistically significant. Previous studies have reported that miR-34a played an important role in the pathogenesis of RA, but the findings were controversial (*Niederer et al., 2012*; *Xie et al., 2019*). Hence, we analyzed the expression of miR-34a-5p in PBMCs using RT-qPCR and found that miR-34a-5p was significantly down-regulated in RA patients compared to health controls (Fig. 11G).

## DISCUSSION

We integrated multiple bioinformatics analysis methods to attain diagnostic genes and vital microRNAs of RA. A total of 14 modules were identified using the WGCNA algorithm. Of these, the turquoise module was large in size (containing 845 genes) and highly correlated with RA, and was selected for further study. We found that the main functions of genes were to participate in leukocyte activation and regulate cell–cell adhesion by performing GO annotation on the turquoise module. KEGG results showed that these genes were highly focused on cytokine-cytokine receptor interaction, chemokine signaling pathway, NF-kappa B signaling pathway, and T cell receptor signaling pathway, in addition to regulating helper T (Th) cell differentiation. Cytokines are small peptides or glycoproteins produced by stimulated immune cells and some stromal cells (e.g., endothelial cells) with a wide range

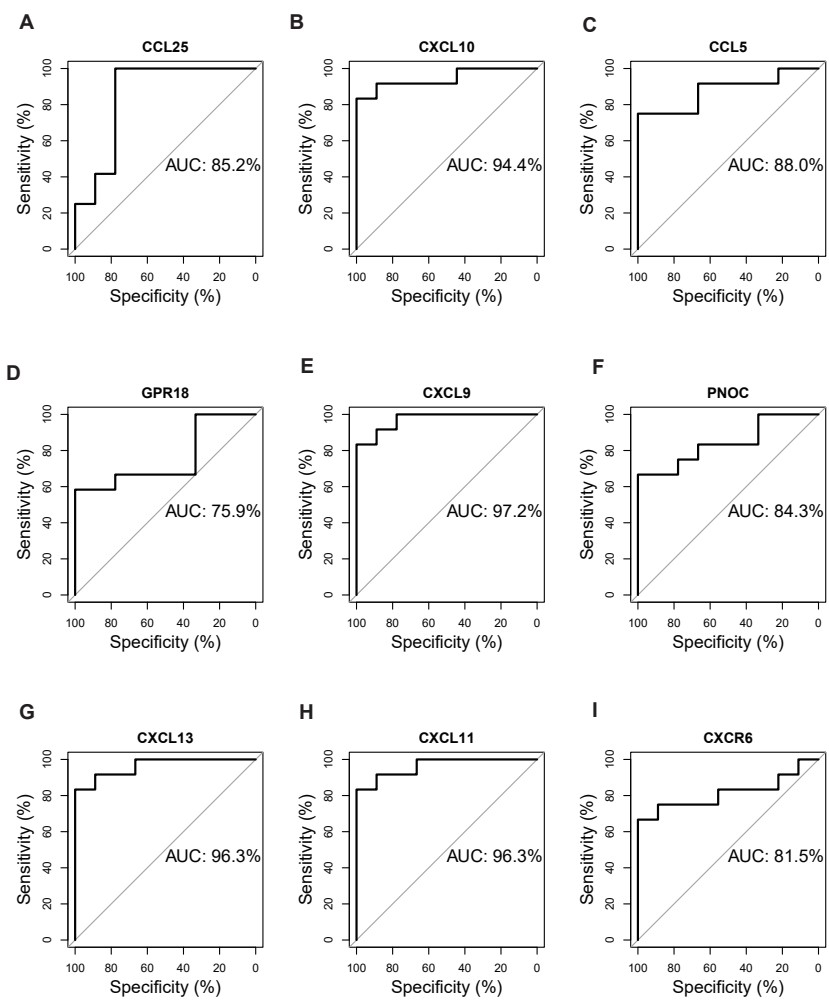

**Figure 9** **Validation of hub genes by ROC curve analysis.** AUC value of each plot was calculated. Genes with AUC values greater than 85% were considered as potential biomarkers for the diagnosis of RA. (A) CCL25, (B) CXCL10, (C) CCL5, (D) GPR18, (E) CXCL9, (F) PNOC, (G) CXCL13, (H) CXCL11, (I) CXCR6.

of biological activities (*Lackie, 2010*). They can mediate long-lasting tissue inflammation and injury by autocrine or paracrine systems and have been shown to be implicated in the pathogenesis of many host autoimmune diseases, such as RA, inflammatory bowel disease, and ankylosing spondylitis (*Schett et al., 2013*). It is believed that TNF and IL-6 are critical pro-inflammatory cytokines that promote osteoclast maturation leading to cartilage degeneration and matrix degradation, and can induce the release of other stimulating factors (such as IL-1), resulting in a vicious cycle of unrelieved synovitis in RA (*Bertolini et al., 1986*; *Ohshima et al., 1998*). The NF-kappa B signaling pathway, activated by the receptor activator of nuclear factor $\kappa$B ligand (RANKL), is also worthy of attention for its pathogenic effect on RA. *Hirayama et al. (2005)*. found that blocking the NF-kappa B pathway with STAT-6 fusion protein significantly alleviated joint inflammation and
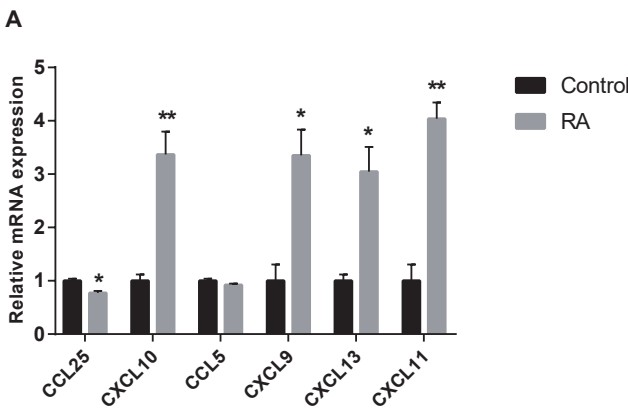

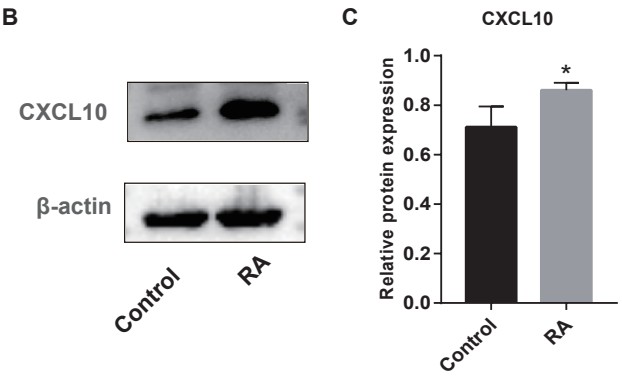

**Figure 10** **Validation of hub genes by RT-qPCR and Western blot.** (A) RT-qPCR validation of the hub gene between RA and normal controls. (B) Expression of CXCL10 in PBMCs was detected with Western blot. (C) The relative expression of CXCL10 was quantified via normalization to $\beta$-actin. * $p$-value $< 0.05$; ** $p$-value $< 0.01$.

bone destruction in mice (*Hirayama et al., 2005*). Moreover, GSEA analysis identified five inflammation-associated gene sets that were up-regulated in the RA phenotype, such as IL6-JAK-STAT3 signaling, interferon $\alpha$/ $\gamma$ response and complement, which were highly consistent with the results of GO and KEGG annotation. The Janus kinase (JAK) family, consisting of four JAK proteins, transduces inflammatory cytokines signals (e.g., IL-6 and interferon) through the JAK-STAT pathway. A double-blind clinical study showed promising clinical improvement in RA patients after oral administration of tofacitinib, a drug that blocks the JAK1-mediated cytokine signaling and inhibits STAT phosphorylation (*Boyle et al., 2015*). We used CIBERSORT to quantify the composition and content of immune cells in synovial tissue to further understand the immune microenvironment of RA. We found that the RA specimens contained higher levels of leukocyte subtypes, such as plasma cells and CD8 T cells, than normal tissue using the Wilcox test. The influx of immune cells into joint forms a complicated inflammatory network that activates fibroblast-like synoviocytes (FLS), inducing synovial vascular proliferation, and recruiting more immune cells to inflammatory sites (*Chen et al., 2019*).

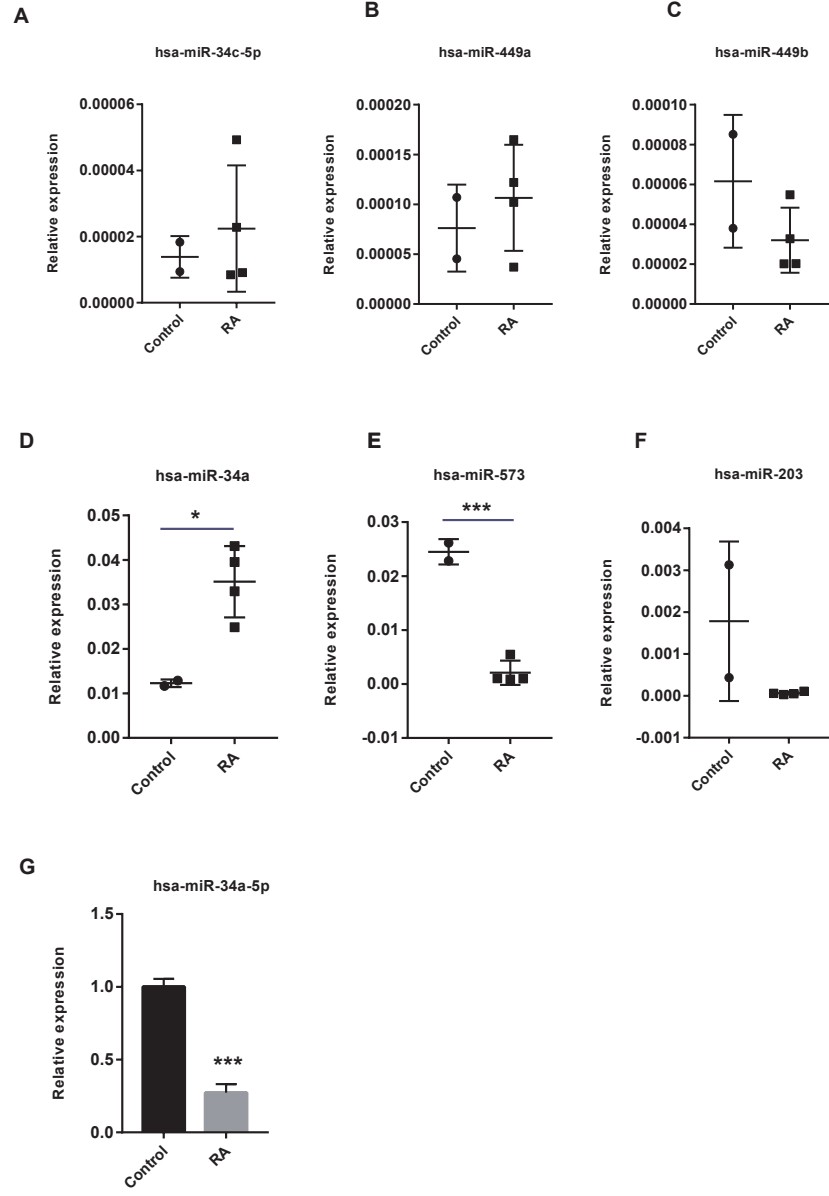

**Figure 11 Validation of vital microRNAs.** (A-F) The relative expressions of hsa-miR-34c-5p, hsa-miR-449a, hsa-miR-449b, hsa-miR-34a, hsa-miR-573, and hsa-miR-203 in synovial tissue. Data were obtained from the GEO dataset GSE37425 and presented as log2 expression of probe intensity (health control: $n = 2$, RA patients: $n = 4$). (G) hsa-miR-34a-5p expression in PBMCs was detected with RT-qPCR. Data were relative to U6 expression. * $p$-value < 0.05, *** $p$-value < 0.001.

We identified five clusters in the PPI network and the most significant one contained nine hub genes, five of which were confirmed by ROC curves analysis and RT-qPCR, including CCL25, CXCL9, CXCL10, CXCL11, and CXCL13. Chemokines can be divided into homeostatic subgroup (e.g., CXCL12) and inflammatory subgroup (such as CXCL9, CXCL10) depending on their function(*Vergunst & Tak, 2005*). However, this functional

classification is not static, and chemokines act in a context-dependent manner based on different tissues and disease stages. A recent study found that plasma levels of chemokines, such as CXCL9, CXCL10, and CXCL13, are much higher in patients with early untreated RA than in normal subjects. Among these discriminators, CXCL10 is associated with indicators of disease activity (e.g., DAS28), which may help determine whether RA patients are in an active phase of the disease (*Pandya et al., 2017*). CXCL10 is mainly produced by interferon-$\gamma$ stimulated T lymphocytes, neutrophils, and monocytes, and is therefore also known as 10 kDa IP-10, which exerts its biological function by binding to the receptor CXCR3 (*Antonelli et al., 2014*). High levels of CXCL10 in humoral components, such as peripheral blood and joint fluid, are typical of various autoimmune diseases, especially those with a predominance of Th1 cells (*Lee, Lee & Song, 2009*). On the one hand, CXCL10 and CXCR3 play an important role in the homing of leukocytes to inflamed tissues (*Romagnani et al., 2002*); on the other hand, the massive accumulation of inflammatory cells, especially Th1 lymphocytes, enhances the secretion of IFN-$\gamma$ and TNF-$\alpha$, which in turn stimulates CXCL10 production, thus amplifying the feedback loop and leading to persistent inflammation and tissue damage (*Antonelli et al., 2008*). Chemokine receptors, which bind to chemokines, have also been shown to be involved in Th1-type and Th2-type inflammatory responses (*Norii et al., 2006*). Specific or non-specific targeting of chemokines and their receptors can be used to treat RA, and many targeted therapeutics have been tested in clinical trials with satisfactory results. For instance, patients taking both methotrexate and MDX1100, an antagonist of CXCL10, achieved reduced disease activity and improved symptom relief (*Yellin et al., 2012*). The application of drugs targeting chemokine receptors, such as CP481,715, showed promising results by reducing the number of macrophages in synovial tissue to ameliorate inflammatory response (*Clucas et al., 2007*).

We obtained 72 miRNA-gene regulatory relationships in our analysis of miRNA-gene interaction, of which seven miRNAs possessed two target genes, including hsa-miR-449a, hsa-miR-297, hsa-miR-203a-3p, hsa-miR-449b-5p, hsa-miR-34c-5p, hsa-miR-573, and hsa-miR-34a-5p. miRNAs belong to non-coding RNAs, which regulate cellular activity by degrading mRNA or inhibiting transcriptional processes (*Esmailzadeh et al., 2017*). It has been reported that about 60% of genes in the human body are regulated by miRNAs, and a single miRNA usually possesses thousands of target genes, which means that abnormal miRNA function may cause pathological consequences such as immune-related diseases and certain types of tumor (*Baltimore et al., 2008*; *O'Connell et al., 2010*). Widely expressed in a variety of immune cells (dendritic cells, monocytes, lymphocytes, etc.), miR-34a plays an important role in regulating cell development, function and survival, which is an important hub in the regulatory network of T cells (*Taheri et al., 2020*). In this study, we found that miR-34a-5p was significantly down-regulated in RA patients compared to health controls. Similarly, it has been shown that miR-34a was not well-expressed in RA synoviocytes, which contributed to their resistance to apoptosis and thus caused persistent inflammation of synovial tissue (*Hou, Wang & Zhang, 2019*; *Niederer et al., 2012*). In addition, *Wu et al. (2021)* found that miR-34a in extracellular vesicles (Evs) was also involved in the development of RA (*Wu et al., 2021*). Elevated expression of miR-34a

in BM-MSC-derived Evs activated the p53 signaling pathway by inhibiting cyclin I (CCNI) expression, which in turn inhibited FLS proliferation and promotes apoptosis, and thus had a positive effect on alleviating the inflammatory state of RA. However, some scholars hold the opposite view. It has been shown that miR-34a expression was up-regulated in PBMCs from RA patients (*Ebrahimiyan et al., 2019*; *Xie et al., 2019*), and silencing miR-34a markedly reduced release of inflammatory mediators (*Kurowska-Stolarska et al., 2017*). In other autoimmune diseases, such as Sjögren Syndrome, the level of miR-34a was also significantly elevated in patients (*Kim et al., 2019*). This controversy suggests that research on miR-34a should continue. miR-449a has recently been identified as an important regulator of RA pathogenesis by suppressing HMGB1 and YY1 expression, thereby attenuating synovial hyperplasia, FLS migration, and the release of inflammatory mediators (*Cai et al., 2019*). *Jin et al. (2018)* found that miR-34c expression was significantly decreased in serum of Treg-depleted mice, an animal model commonly used in studies of autoimmune diseases (*Jin et al., 2018*).

Our study has limitations, including our focus on the most critical turquoise model in the process of co-expression network analysis, without adequate discussion of the black, blue, and purple modules, which were also strongly associated with RA. Our study size was small, and in-depth clinical studies are needed to support our findings.

## CONCLUSIONS

In summary, we identified diagnostic genes and vital microRNAs associated with RA, which can help us to better understand the pathogenesis behind this disorder and provide theoretical support for exploring more effective therapies.

### Funding

The present study was supported by National Natural Science Foundation of China (grant number 81904053), the Plan Project of Hangzhou Health Science and Technology Department (grant number 2018B028), and the Opening Project of Zhejiang Provincial Preponderant and Characteristic Subject of Key University, Zhejiang Chinese Medical University (grant number ZYX2018008). The funders had no role in study design, data collection and analysis, decision to publish, or preparation of the manuscript.

### Grant Disclosures

The following grant information was disclosed by the authors:
National Natural Science Foundation of China: 81904053.
Hangzhou Health Science and Technology Department: 2018B028.
Zhejiang Provincial Preponderant and Characteristic Subject of Key University, Zhejiang Chinese Medical University: ZYX2018008.

### Competing Interests

The authors declare there are no competing interests.

## Author Contributions

- Conglin Ren conceived and designed the experiments, performed the experiments, authored or reviewed drafts of the paper, and approved the final draft.
- Mingshuang Li performed the experiments, prepared figures and/or tables, and approved the final draft.
- Yang Zheng, Fengqing Wu and Weibin Du analyzed the data, prepared figures and/or tables, and approved the final draft.
- Renfu Quan conceived and designed the experiments, authored or reviewed drafts of the paper, and approved the final draft.

## Human Ethics

The following information was supplied relating to ethical approvals (i.e., approving body and any reference numbers):

This study was approved by Ethics Committees of Xiaoshan Traditional Chinese Medicine Hospital (Ethical Application Ref: 2020012).

## Data Availability

The R codes and raw data are available in the Supplemental Files.

## Supplemental Information

Supplemental information for this article can be found online at http://dx.doi.org/10.7717/peerj.11427#supplemental-information.

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
