# Peer review of "Identification of diagnostic genes and vital microRNAs involved in rheumatoid arthritis: based on data mining and experimental verification"

_PeerJ, doi:10.7717/peerj.11427_

## Round 0.1 · original submission · Major Revisions

Dear Dr. Ren,

Your manuscript has been reviewed by two reviewers. Reviewers have some major concerns regarding the experimental design especially reviewer 1. Please check the comments and revise the manuscript accordingly.

·

Basic reporting

English language is a concern throughout the manuscript. Another point is consistency of writing and use of terminology.
References aren't appropriately used.
Introduction is not well connected.

Experimental design

No original research.
No gene enrichment analysis was performed.
Research question was not well defined.
There is no rigorous investigation performed.

Validity of the findings

No novelty, this type of reports already published. (https://www.ncbi.nlm.nih.gov/pmc/articles/PMC6657850/).
Results were not interpreted and explanatory.
This is just use of bioinformatic tools study. Author only used qRT PCR to validate the genes but not western blots for their protein products.

·

Basic reporting

The manuscript titled "Identification of diagnostic genes and vital
microRNAs involved in rheumatoid arthritis: based on
data mining and experimental verification" has been written well. There are very few grammatical errors in the manuscript with sufficient literature review provided. The data analysis and representation is adequate. The authors have missed to add the findings of a) Clin Exp Rheumatol. Jul-Aug 2017;35(4):586-592. Epub 2017 Jan 27.; b) Zhu X, Wu L, Mo X, et al. Identification of PBMC-expressed miRNAs for rheumatoid arthritis. Epigenetics. 2020;15(4):386-397. doi:10.1080/15592294.2019.1676613. The authors should consider including them in the discussion.

Experimental design

The experimental design is well planned to address the hypothesis with well defined objectives. There is adequate information for the repetition of the experiment by others.

Validity of the findings

The findings are novel for the biomarkers although there are reports on the miRNAs involved in RA patients. The conclusions are well supported with the findings of the study. The authors should validate a few of the miRNAs for functional role in regulating the inflammatory genes mentioned in the study as a functional validation of the study.
The authors have mentioend that the highly altered pathways belong to inflammation and cell-cell adhesion but they have not validated any gene for the cell-cell adhesion pathway or looked into its miRNA regulation. Why?

Additional comments

The authors should rectify the grammatical errors and space between the words in the manuscript.

---

## Round 0.2 · Major Revisions

Dear Dr. Ren,

The manuscript has been reviewed by two experts in the field. While one reviewer has accepted the manuscript, the other reviewer has raised significant concerns that need to be addressed. Please look at the suggestions pointed out by reviewer 2 and make the changes in the manuscript accordignly.

·

Basic reporting

Clear and unambiguous and language is appropriate.
Sufficient field background and references
Figures and structured articles

Experimental design

Articles fit in Aims and scope of the Journal
Research question is improved in revised manuscript.
Investigation in details.
Methods described.

Validity of the findings

Findings are well defined. All data have been provided. Conclusions are well stated.

Additional comments

Author address my concerns.

Reviewer 3 ·

Basic reporting

The language can be made better

Experimental design

The question is well defined. The GSEA has made this manuscript better.

Validity of the findings

Conclusions could have been better stated.

Additional comments

This reviewer is of the opinion that the authors have done a good job with the rebuttal. However, though not fully agreeing with the other reviewer's comments about the research not being entirely novel, I feel that the conclusions of this manuscript can be made much better.
1) The strength of this manuscript is in the microRNAs prediction. Hence, I would suggest the authors to expand greatly on the ROC curve prediction and microRNA validation.
2) I would highly recommend running a few Western blots for protein validation
3) Please expand the discussion to put more focus on the microRNA validation and the results from the Western Blots (if run) from point 2

---

## Round 0.3 · accepted · Accept

Dear Dr. Ren,

The manuscript is now been accepted.

Reviewer 3 ·

Basic reporting

Good

Experimental design

Good

Validity of the findings

Novel. Good

Additional comments

Congratulations! The authorshave satisfactorily answered all my concerns. The manuscript is very good and should be published.